# How Long Does It Take to Regain Normocalcaemia in the Event of Postsurgical Hypoparathyroidism? A Detailed Time Course Analysis

**DOI:** 10.3390/jcm11113202

**Published:** 2022-06-03

**Authors:** Laura Guglielmetti, Sina Schmidt, Mirjam Busch, Joachim Wagner, Ali Naddaf, Barbara Leitner, Simone Harsch, Andreas Zielke, Constantin Smaxwil

**Affiliations:** 1Department of Surgery, Kantonsspital Winterthur, 8400 Winterthur, Switzerland; laura.guglielmetti@ksw.ch (L.G.); sina.schmidt@ksw.ch (S.S.); 2Department of Endocrine Surgery, Endocrine Center Stuttgart, Diakonie-Klinikum Stuttgart, 70176 Stuttgart, Germany; buschm@diak-stuttgart.de (M.B.); wagner@diak-stuttgart.de (J.W.); naddaf@diak-stuttgart.de (A.N.); andreas.zielke@diak-stuttgart.de (A.Z.); 3Outcomes Research Unit, Endocrine Center Stuttgart, Diakonie-Klinikum Stuttgart, 70176 Stuttgard, Germany; leitner-barbara@t-online.de (B.L.); simone.harsch@diak-stuttgart.de (S.H.)

**Keywords:** thyroid surgery, postsurgical hypoparathyroidism, risk-factor analysis, time course

## Abstract

Background: Postsurgical hypoparathyroidism (PH) is the most common side effect of bilateral thyroid resections. Data regarding the time course of recovery from PH are currently unavailable. Therefore, a detailed analysis of the time course of PH recovery and conditions associated with rapid recovery was conducted. Methods: This is a retrospective analysis of prospectively documented data. Patients with biochemical signs of PH or need for calcium supplementation were followed-up for 12 months. Logistic regression analyses were used to identify covariates of early as opposed to late recovery from PH. Results: There were 1097 thyroid resections performed from 06/2015 to 07/2016 with n = 143 PH. Median recovery time was 8 weeks and six patients (1.1% of total thyroid resections) required calcium supplementation > 12 months. Recovery of PH within 4 and 12 weeks was characterized by high PTH levels on the first postoperative day (4 weeks: OR 1.13, 95% CI 1.06–1.20; 12 weeks: OR 1.08, 95%CI 1.01–1.16). Visualization of all PTGs emerged as an independent predictor of recovery within 12 months (OR 2.32, 95% CI 1.01–4.93) and 24 weeks (OR 2.69, 95% CI 1.08–6.69). Conclusion: In the setting of specialized high-volume endocrine surgery, permanent PH is rare. However, every second patient will require more than 2 months of continued medical surveillance. Early recovery was associated with only moderately decreased postsurgical PTH-levels. Successful late recovery appeared to be associated with the number of parathyroid glands visualized during surgery.

## 1. Introduction

Postsurgical hypoparathyroidism (PH) is the most common complication following total thyroidectomy [1]. PH impairs a patient’s convalescence and prolongs outpatient follow-up [2,3]. Patients experiencing PH require increased medical attention, repeated biochemical assessment, as well as additional medication, all of which affect medical resource use. Moreover, patients with severe, persisting PH often suffer incapacitating symptoms and are likely to be confronted with a reduced quality of life, as well as reduced life expectancy [4].

At present, no consensus exists for the definition of PH, nor for the interval until PH might have to be considered a permanent condition. Traditional cut-offs for permanent PH are either 6 [5,6,7,8,9,10,11,12] or 12 months [1,2,13]. Most authors use biochemical definitions for PH, such as PTH levels (with different cut-offs) [1] or postoperative hypocalcemia [12,13,14,15]. Less frequently, newly prescribed postoperative calcium and/or vitamin D supplementation are taken as a surrogate to characterize the event of PH as a hormonal insufficiency state.

However, data regarding the course of convalescence after PH are scarce, and to the best of our knowledge, a detailed analysis of individual time courses of recovery from PH, as well as the clinical and biochemical data associated with the length of the period of convalescence are currently not available. Such insight would be highly desirable to allow for the optimum allocation of resources. Determining the expected time course of individual PH recovery could facilitate identifying patients who are dependent on prolonged and experienced medical support and, therefore, may benefit from specialized medical professionals.

Therefore, this study conducted a detailed analysis of the biochemical recovery of patients with PH. Moreover, this study aimed to identify predictors for early recovery (i.e., 4 weeks), as well as for prolonged (i.e., more than 24 and 52 weeks) postoperative hypocalcemia.

## 2. Materials and Methods

### 2.1. Study Population

We performed a selective analysis of prospectively documented data of a consecutive series of patients undergoing thyroid surgeries at a specialized centre of Endocrine Surgery. Datasets of the registry, medical charts, as well as data from the Centres’ structured follow-up program were analysed including all patients undergoing thyroid surgeries in this institution between June 2015 and May 2016. With regard to the datasets until discharge of an individual patient, there were no missing values. Patients with co-existing parathyroid pathology diagnosed either preoperatively or incidentally during surgery for a thyroid pathology were excluded from this analysis (n = 64 of which n = 23 total thyroidectomy, n = 26 hemithyroidectomies, and n = 15 other procedures).

All individuals in this series underwent surgery at a high-volume centre of reference in thyroid and parathyroid surgery, certified by the German Association of Surgery (DGAV) [16]. All operations were attended by a certified Endocrine Surgeon, using highly standardized surgical protocols with a focus to apply minimal invasive techniques whenever possible. These techniques do not mandate extensive dissection to visualize all four parathyroid glands, but encourage the demonstration and preservation of at least one parathyroid gland during the procedure. Moreover, parathyroid glands that were attached to the thyroid lobe and could not be preserved in situ were routinely dissected and subject to auto-transplantation into the ipsilateral sternocleidoid muscle. In accordance with current practice guidelines, conventional medical therapy of PH always involved the concomitant use of calcium supplements and active vitamin D preparation (i.e., Calcitriol; 1.25 (OH)2D3; 0.25–0.5 µg bid).

### 2.2. Postsurgical Follow-up Data

All patients in this series were subject to a standardized perioperative recovery protocol, ensuring biochemical testing for fasting (total serum) calcium, and PTH levels between 7 and 8 a.m. on post-operative day (POD) 1 and POD2, as well as POD3 in the rare event a patient had not yet been discharged. Calcium and PTH levels were determined using a Cobas pro (Cobas e 801) from Roche (ECLIA, intact PTH: normal range 15 to 65 pg/mL; total serum calcium normal range 2.2–2.6 mmol/L).

All patients with either symptoms of PH or biochemically proven hypocalcaemia and/or hypoprothrombinaemia or newly prescribed calcium plus vitamin D (i.e., 1,25 (OH)2D3) medication were defined as potential candidates for PH and subjected to a detailed follow-up program. To this end, the Endocrine Centers Outcomes Research Unit used regular (i.e., monthly) structured telephone interviews to assess current medication (specifically: calcium supplements and vitamin D3), laboratory parameters, and symptoms of hypocalcemia. Follow-up was terminated only if a full recovery had been validated by the biochemical demonstration of normocalcaemia, as well as euparathormonaemia (as determined by the respective local assay or test) and the individual was without oral calcium intake. For the purpose of this paper, 12-month results were analysed. In the event a patient was “lost to follow-up” after multiple unsuccessful calls, the last observation was carried on forward and the outcome documented accordingly (LOCF).

### 2.3. Definition of Postsurgical Hypoparathyroidism

Historically, the duration of PH is termed and classified taking account of two different intervals: transient PH, i.e., PH lasting for less than 12 or less than 24 weeks, and permanent PH for cases lasting longer than either 6 or 12 months. This is also found in most clinical guidelines [1,5,6,7,8,9,10,13,17,18]. Hence, predictors of the duration of PH were determined using logistic regression analysis at these specific time points: 12, 24, and more than 24 weeks after surgery. In accordance with current guidelines, all cases with clinical signs of hypocalcaemia or need for calcium and active vitamin D (i.e., Calcitirol; 1,25 (OH)2D3) medication or biochemically proven hypocalcaemia and/or hypoprothrombinaemia persisting for more than twelve months after surgery were defined to have “permanent hypoparathyroidism” [1,5,6,7,8,9,10,13,17,18].

### 2.4. Handling of Data and Statistical Analysis

All perioperative data were prospectively documented as individual datasets according to the thyroid-surgery-module of the StuDoQ-Quality Assurance Registry of the German Surgical Association (DGAV) after informed consent [16]. All data computed for this publication were pseudonymized or aggregate non-individual data. Descriptive statistics was used to summarize patients’ characteristics. Continuous variables are reported as the mean and standard deviation (SD) or median and interquartile range and were compared between groups using two-sample independent t-tests or Mann–Whitney U-tests (non-normal data). Categorical variables were summarized as frequencies (%) and compared using Pearson’s chi-squared test or Fisher’s exact test where applicable.

The variables included for univariate analysis in the logistic regression models were a priori determined after literature review. A total of n = 8 variables were tested in univariable models: gender, presence of Graves’ disease, malignant thyroid conditions, visualization of all parathyroid glands during thyroid surgery, reimplantation of parathyroid glands, PTH and calcium on POD 1, and symptoms of hypocalcemia. Variables with a *p*-value of <0.15 were retained in the multivariable models. All variables retained in the multivariable models had no more than a weak correlation (Spearman correlation coefficient r < 0.39, as suggested by Evans et al.) [19]. In this cohort of patients, there was no strong correlation between female gender and presence of graves’ disease (Spearman correlation coefficient r = 0.147, *p* = 0.079).

Results are reported as the odds ratio (OR) with the corresponding 95% confidence interval (CI). Goodness of fit was tested using the Hosmer–Lemeshow test, and the area under the curve (AUC) is reported for the predictive accuracy of the model. SPSS version 25 (IBM corp., Armonk, NY, USA) and R Studio Version 3.2.1. (RStudio, Inc., Boston, MA, USA) were used for data analysis. *p*-values < 0.05 (two- tailed) were considered statistically significant.

### 2.5. Ethics Approval and Consent to Participate

This study was approved by the Institutional Review Board of the Diakonie-Klinikum Stuttgart and conducted in cooperation with the Endocrine Centers certified Outcomes Research Unit. All methods were carried out in accordance with the Declaration of Helsinki and the approved guidelines. As mentioned, data enrolled in this study were obtained from the StuDoQ quality assurance database, for which informed consent had been obtained. The data presented in this article do not represent a human participant research study and do not include personal identifying information. This secondary analysis was carried out for the purpose of quality assurance and did not require informed consent.

## 3. Results

During the 12-month period, a total of 1097 + 69 surgeries for thyroid pathologies were performed. There were 69 cases that had to be excluded because of co-existing parathyroid pathologies with removal of at least one parathyroid gland. Thyroid resections comprised 538 complete resections (505 total thyroidectomies (TTX) and 33 two-staged thyroidectomies) and 559 less-than-total resections (515 hemithyroidectomies and 44 comprising of central resections, as well as bilateral subtotal resections such as Hartley, Dunhill, Enderlein–Hotz). The patient characteristics of all procedures are displayed in Table 1. Of all thyroid patients, 74.5% were female, and 85.9% had benign conditions including 86 patients with Graves’ disease, all of which had total thyroidectomies (7.8% of the entire cohort, 15.9% of TTX). A total of n = 155 patients had malignant thyroid pathologies (14.1% of the entire cohort, 27.9% of TTX), of which 67 patients had a systematic dissection of cervical lymph nodes involving Robbins-Regions VI and VII (6.1% of the cohort, 12.8% of TTX). There were 126 patients with previous surgery to the neck (rate of reoperations: 11.5%). Surgery for recurrent benign goitre was recorded in 54 cases (4.9%). Of these, nine had bilateral and the remainder unilateral reoperations (83.3%).

There were 154 patients with either newly prescribed calcium and vitamin D (i.e., Calcitriol; 1,25 (OH)2D3) medication or hypocalcaemia or hypoprothrombinaemia or hypocalcaemic symptoms on any postoperative day. Of these, 11 had to be excluded because of incomplete data (n = 2) or because they were not willing to be entered into the follow-up program (n = 9), leaving 143 patients for analysis (92%).

### 3.1. Postsurgical Hypocalcaemia—Patient Characteristics

The majority of patients with PH were female n = 122 (88.3%); n = 127 (88.8%) had undergone total thyroidectomy (see Table 1 and Figure 1), and n = 14 (9.8%) patients had a two-stage thyroidectomy. One patient (0.7%) had unilateral hemithyroidectomy plus subtotal resection of the contralateral lobe, while another patient (0.7%) had a unilateral hemithyroidectomy and no history of thyroid surgery in the past. Both of them had lowered PTH levels and symptoms of hypocalcaemia on POD1.

Histopathological diagnoses confirmed benign thyroid conditions in 123 and included 16 patients with Graves’ disease (11.2%) and 12 redo-operations (8.4%). There were 20 patients confirmed to have thyroid cancer (14%), and a dissection of the central cervical lymph nodes had been carried out in 22 patients (16.8%).

Parathyroid gland visualization was recorded in all of the cases, with one gland documented in three (3.1%), two glands in 22 (15.4%), three glands in 43 (30.1%), and four glands in 68 patients (47.5%), respectively. There were seven patients in whom no parathyroid gland had been visualized (4.9%). Parathyroid gland auto-transplantation was recorded in n = 33 (23.1%) patients and involved one gland in n = 31 (94%) and two glands in n = 2 (6%). These data were not different from the entire cohort of bilateral procedures (Table 1).

Of 143 patients, 81 (56.6%) reported symptoms of hypocalcaemia on any given day during the postoperative course (Figure 2). Likewise, 63 (44.1%) had calcium of less than 2.0 mmol/l (normal range 2.0–2.75 mmol/l), of which n = 23 had calcium of less than 2.0 mmol/l on POD 1. Of all patients, 114 (79.7%) had a PTH level below the lower limit of norm (iPTH-assay range 15–65 pg/mL), and of these, 109 had a PTH of <15 pg/mL recorded on POD 1. Calcium glucoronate starting at the same day of the surgical procedure had been ordered in 15 cases (10.5%). Of these cases, n = 2 had undergone hemithyroidectomy because of recurrent benign nodular goitre, n = 3 had a total thyroidectomy with central lymph node clearance for thyroid cancer, n = 2 had a total thyroidectomy for Graves’ disease, and n = 4 had at least one autografted parathyroid gland. Symptoms of hypocalcemia, albeit with a normal biochemical parameter, had been recorded in n = 6. However, taking account of the entire period of postoperative hospitalization, all but two patients (n = 141) had been ordered postoperative calcium supplementation until discharge. Two patients chose not to take supplementary calcium medication despite symptoms of hypocalcaemia.

### 3.2. Postsurgical Hypoparathyroidism—Time Course of Recovery

Median recovery time of PH for all patients was 8 weeks, and two thirds of all patients recovered within 18 weeks (IQR 4–18 weeks). As depicted in Figure 2, the slope of the curve of recovery was steepest and displayed almost linear degression in the first and second 4-week intervals post-surgery. During this period, 50% of patients had a biochemical recovery, whereas the next 25% of patients took more than twice that long for full recovery from PH. Of the 15 patients that had been ordered calcium and vitamin D3 to be initiated immediately after surgery and prior to any biochemical analysis, all had biochemical signs of hypoparathyroidism, and all of them recovered (1 to 33 weeks). Biochemical recovery could not be proven in 3 three patients who were free of symptoms and off calcium and vitamin D, but had self-terminated the follow-up after a median of 16 weeks (IQR 15–17) without any further laboratory testing. Only one patient (0.7% of the cohort, 0.2% of TTX and 0.09% of all thyroid surgeries) required calcium substitution for more than 12 months (Figure 3).

Historically and currently presented in the respective guidelines, PH is termed transient whenever full recovery occurs either within 12 or within 24 weeks, and it is considered persistent if a biochemical recovery is not documented at 24 weeks [1,5,6,7,8,9,10,17] or after a period of 12 months [13,18]. In order to interpret potential predictors for the duration or recovery of PH, these intervals were, therefore, taken into account.

Baseline characteristics of patients with a documented recovery within 12 weeks as compared to more than 12 weeks are summarized in Table 2, and those for recovery within 24 weeks as compared to a later recovery are presented in Table 3. Median PTH on POD1 was significantly higher for patients with earlier recovery of PH in each of these groups. Patients with a full recovery within 12 weeks showed a higher PTH level on POD1 than those that took longer (median 11.1 (IQR 7.3–11.6) vs. 9.1 (IQR 9–15.4), *p* = 0.006). Likewise, PTH levels were higher in those individuals in whom full recovery was documented within 24 weeks as opposed to longer than 24 weeks to recovery (median 10.8 (8.45–15.1) vs. 8.8 (7.3–12.7), *p* = 0.003). Of note, there was a steady decline of POD1 PTH levels when computed against time to recovery, suggesting that besides predicting the incident, PTH levels (or percentage decline as compared to presurgical levels) may allow for an estimation of individual recovery times (Figure 4).

Calcium levels on POD1 or POD2, number of patients with hypocalcaemic symptoms, and percentage of patients undergoing total thyroidectomy per group were not statistically different (Figure 5). This may in part be due to the early and frequent use of perioperative calcium and vitamin D3 substitution in this study.

### 3.3. Postsurgical Hypoparathyroidism—Predictors for Time to Recovery

In order to further explore the utility of clinicopathological variables to predict time to recovery, patients were stratified according to time to recovery and potential predictors of recovery assessed using logistic regression analysis as described above.

The presence of graves’ disease, thyroid cancer diagnosis, visualization of the number of parathyroid glands, replantation of parathyroid glands, PTH, and calcium level on POD 1, as well as symptoms of hypocalcemia were tested by univariate analysis and retained in the model in the case of a *p* < 0.015.

A total of n = 41 patients (28.7%) recovered within 4 weeks after surgery. PTH on POD1 emerged as a significant predictor for early recovery (OR 1.13 (95% CI 1.06–1.2) *p* < 0.001) when controlled for Graves’ disease and visualization of all PGs (Table 3).

Transient PH, defined as recovery within 12 weeks after surgery, was recorded in n = 96 patients. Visualization of all PGs (OR 2.3 (95% CI 1.01–4.93), *p* = 0.029) and PTH on POD1 (OR 1.08 (95% CI 1.01–1.16), *p* = 0.034) were significant predictors for recovery within three months (Table 4).

Within 24 weeks following surgery. n = 116 patients had recovered from PH, while the remaining n = 27 patients remained on calcium supplementation for 25–56 weeks, and n = 3 patients were on continued calcium supplementation (i.e., did not recover) after 12-month follow-up. Univariable and multivariable logistic regression analysis revealed visualization of all PGs as a significant predictor of recovery within 24 weeks (OR 2.69 (95% CI 1.08–6.69), *p* = 0.033) when controlled for PTH levels on POD1. While PTH on POD 1 evolved as a predictor of early recovery of PH, later recovery was associated with visualization of all PTGs during surgery when controlled for Graves’ disease and female gender.

## 4. Discussion

Postoperative hypocalcaemia due to impaired parathyroid function is the most common complication after total thyroidectomy. Although much less frequent, permanent hypoparathyroidism imposes an important medical burden—due to the need for additional medication, regular medical visits, and thus, also significant long-term costs [20]. There is still no consensus with regard to the definition of postoperative hypoparathyroidism (PH), nor the time period after which PH may have to be considered a permanent condition.

Much of the uncertainty is due to a lack of data regarding potential causes of PH, risk factors of PH, as well as the time-course and likelihood of reconvalescence from PH. For instance, a recent “raise-your-hand survey” carried out during the British Association of Endocrine and Thyroid Surgeons (BAETS) meeting in Barcelona revealed only some 10% of the attending 220 surgeons to know their rate of inadvertent parathyroidectomy [21].

However, most guidelines use cut-offs for permanent PH of either 6 months [5,6,7,8,9,10,11,12] or 12 months [1,2,13]. Clinical studies use biochemical definitions for PH such as PTH levels, albeit a variety of different cut-offs, or (total) calcium levels, again using a range of cut-off levels. Moreover, by virtue of the variety of PH definitions, the published rate of symptomatic hypocalcaemia varies between 0.1% and 20.2%, and biochemical hypocalcaemia is reported to occur at a rate of 15–75% [12,13,14,15,22]. Despite these variations, a recent review found no difference of the incidence of PH at 6 versus 12 months (4.11% and 4.08%) [23].

Some studies have used more inclusive definitions for PH [2,3], arguing that postoperative supplementation of calcium and/or newly started active vitamin D (Calcitriol; 1,25 (OH)2D3) medication affect biochemical and clinical findings. On the basis of this inclusive definition of PH, we found the overall incidence in this consecutive series of patients undergoing bilateral thyroid procedures for a wide array of thyroid pathologies to be 24.5% in bilateral procedures. While almost all of the affected patients had at least one level of total serum calcium below the range of normal (98.6%), symptomatic hypocalcemia was less frequent, occurring in 57% of patients with PH, i.e., 13.7% of bilateral procedures.

Overall, we determined the recovery rate of PH at 6 and 12 months to be 82% and 98%, and there was only one patient who did not fully recover within 12 months (0.7%). Bilezikian et al. reported rates for PH at 24 weeks ranging between 0.9% and 1.6% in centres with a record of excellence in endocrine surgery and suggested the 6-month time period to be sufficient to ascertain permanent postoperative hypoparathyroidism [6]. Others have published rates of permanent PH ranging from 0.5% to 8.6% [1,2,10,11,13] and rates of recovery ranging from 6.9% to 46%, again using a broad range of definitions [6,17]. However, besides such “endpoint-observations” at either 6 or 12 months, the current literature does not offer any insight into the actual time course of recovery from PH.

For the first time, we are now able to report a median recovery time for PH of 8 weeks (IQR 4–18). Almost two thirds of patients with PH had a documented recovery within 12 weeks, and some 8% of all patients with bilateral thyroid procedures required continued medical surveillance for more than 12 weeks. However, some 15% of patients with PH, i.e., 3.5% of all patients with bilateral thyroid procedures, take longer than 6 months to recover.

With regard to prognostic factors of recovery, one of the first prospective studies to suggest postoperative PTH levels, specifically the magnitude of decrease, to be a significant predictor of time to recovery was published by Al Dhahri et al. Their study included n = 53 patients undergoing total thyroidectomy. The authors found 15 (28.5%) of their patients to have PH [1], comparing well to the rate in this current study. A large systematic review of 69 studies did, however, not confirm a particular PTH threshold, neither as a predictor of the prevalence, nor the duration of PH [24]. A recent retrospective study using predefined PTH thresholds suggested the combination of postoperative PTH plus calcium levels to be better predictors of the likelihood of PH [25].

In the present study, median PTH levels on POD1 were significantly higher in patients with earlier recovery, and PTH on POD1 emerged as a significant predictor for early recovery within 4 weeks, as well as for recovery within 12 weeks. Moreover, we determined an increasing probability of an earlier recovery with each increment of PTH on POD1. For patients in whom we had observed longer recovery, e.g., more than 12 weeks, intraoperative visualization of (all) parathyroid glands (PGs) was significantly associated with recovery and remained the only significant predictor of recovery for patients with a recovery period of 24 weeks. Although these observations clearly suggest the intraoperative visualization of PGs to be linked to a late likelihood of recovery, we believe that extensive dissection to demonstrate PGs during surgery is not warranted. Indeed, as had already been emphasized by others, this carries the potential to increase the rate of patients experiencing PH (13,23). It has, however, recently been suggested that taking account of the number of PGs in situ together with measurements calcium and PTH during the first month post thyroidectomy may allow predicting a swift parathyroid recovery [13,21,26,27].

Clearly, one of the limitations of this work is the retrospective character. Because this is a representation of a real-world cohort, data regarding preoperative calcium levels, corrected or ionized calcium values, preoperative medication, or preoperative vitamin D levels were not consistent enough to allow for proper evaluation. This might have been worthwhile, since preoperative vitamin D deficiency has been identified as a possible risk factor of PH [28]. Moreover, as a result of the inclusion criteria of the present study, there is a lack of data from “controls”, so no conclusion can be drawn with regard to the association of postoperative PTH levels with the incidence of PH.

However, this is the first report of a detailed time course analysis of recovery from PH in a larger cohort of patients with this condition. Based on the findings, a 6-month cut-off to ascertain permanent PH appears to be inappropriate. However, the majority of patients will recover from PH within 8–12 weeks. The time course detailed in this study should allow for better counselling and planning of out-patient treatment.

Nevertheless, postoperative hypoparathyroidism continues to be a challenge, and it remains to be shown to what extent new techniques such as autofluorescence and fluorescence angiography of the parathyroid glands may lead to an improvement.

## 5. Conclusions

Permanent hypoparathyroidism is a rare event in the hands of specialized high-volume endocrine surgeons. Transient hypoparathyroidism is, however, observed in every fourth patient with a total thyroidectomy. In every second patient, restoration of normocalcaemia and euparathormonaemia required more than 2 months of continued medical surveillance. Early recovery was more likely if postsurgical PTH levels were only slightly decreased. When recovery took longer, full recovery appeared to be associated with the number of parathyroid glands visualized during surgery.

## Figures and Tables

**Figure 1 jcm-11-03202-f001:**
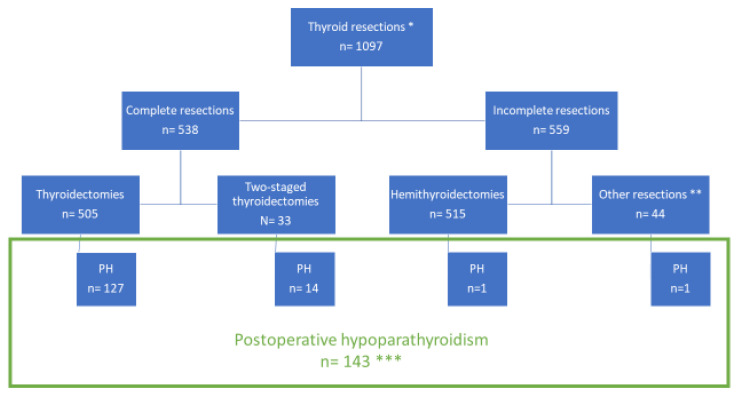
**Overview of the extent of resection in patients included into the study.** * There were that 69 had to be excluded due to co-existing parathyroid pathologies with removal of at least one parathyroid gland, leaving 1097 for analysis; ** comprising central resections, as well as bilateral subtotal resections such as Hartley, Dunhill, Enderlein–Hotz; *** patients with complete datasets.

**Figure 2 jcm-11-03202-f002:**
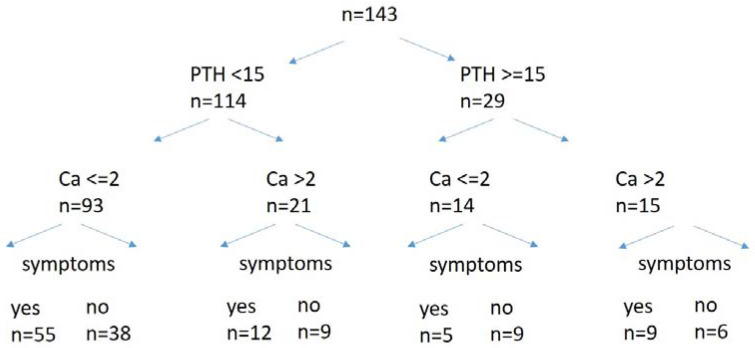
**Overview of biochemical and clinical findings in patients with postsurgical hypoparathyroidism.** Roche Cobas pro e 801; intact PTH = parathyroid hormone (ECLIA; 15 to 65 pg/mL), Ca = total serum calcium (2.2–2.6 mmol/L).

**Figure 3 jcm-11-03202-f003:**
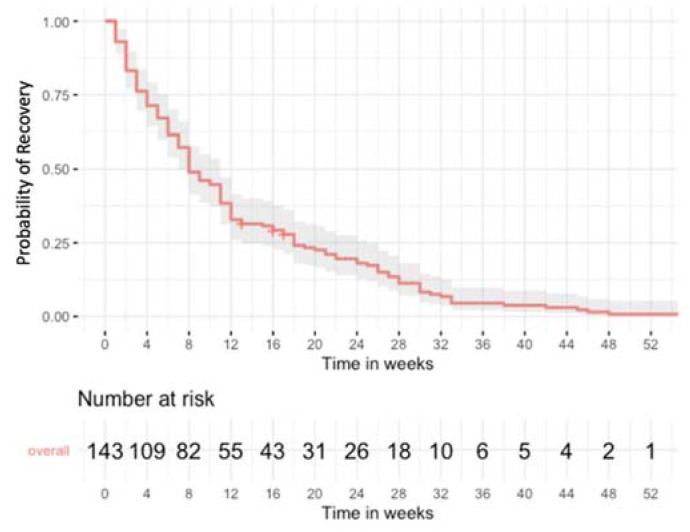
Time to recovery for the entire cohort of patients with postsurgical hypoparathyroidism (Kaplan–Meier).

**Figure 4 jcm-11-03202-f004:**
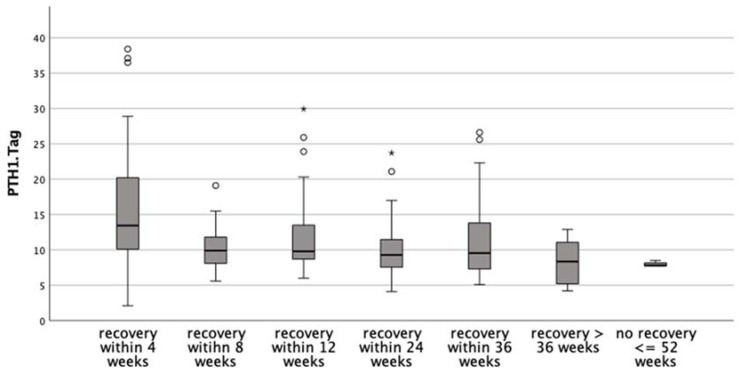
PTH—level on the first postoperative day and time of biochemical recovery from postsurgical hypoparathyroidism. * extreme value, ° outlier.

**Figure 5 jcm-11-03202-f005:**
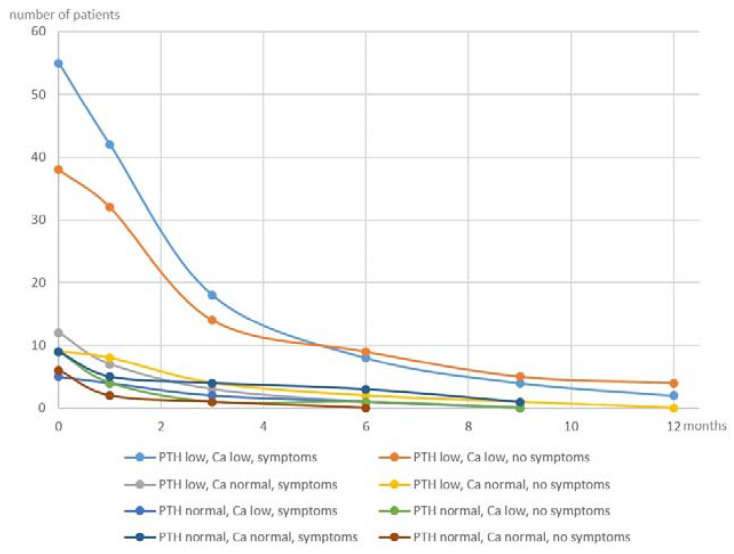
Time to recovery from postsurgical hypoparathyroidism—depicted according to the eight groups shown in Figure 2.

**Table 1 jcm-11-03202-t001:** Baseline characteristics for all patients and all patients with postsurgical hypoparathyroidism.

n (%) Unless Otherwise Stated	All Procedures	Procedures w Postsurgical Hypoparathyroidism (PH) n = 143
	n = 1097	
Malignant thyroid condition	155 (14.1)	20 (14)
Central LAD	67 (6.1)	22 (16.8)
Graves’ disease	86 (7.8)	16 (11.2)
Reoperative Surgery	126 (11.5)	12 (8.4)
Thyroidectomy	505 (46)	127 (88.8)
Hemithyroidectomy (HTx)	515 (46.9)	2 (1.4)
Two-staged bilateral HTx	33 (3.0)	14 (9.8)
PTG visualised		
0	71 (6.5)	7 (4.9)
1	205 (18.7)	3 (3.1)
2	390 (35.6)	22 (15.4)
3	152 (13.9)	43 (30.1)
4	279 (25.4)	68 (47.5)
PTG autograft		
0	938 (85.5)	110 (76.9)
1	146 (13.3)	31 (21.7)
2	12 (1.1)	2 (1.4)
4	1 (0.1)	0
PTH POD 1, median (IQR)	-	10.4 (8.1–14.4)
PTH POD 2, median (IQR)	-	9.8 (7.6–11.9)
Calcium POD 1, median (IQR)	-	2.1 (2.0–2.2)
Calcium POD 2, median (IQR)	-	2.0 (1.9–2.1)

Abbreviations: LAD = lymphadenectomy, POD = postoperative day, PTG = parathyroid glands, PTH = parathyroid hormone; IQR = interquartile range.

**Table 2 jcm-11-03202-t002:** Baseline characteristics for recovery within 12 weeks compared to a later recovery.

n (%) Unless Otherwise Stated	Recovered ≤ 12 Weeks	Recovered > 12 Weeks	*p*-Value
	n = 96	n = 47	
Female gender	85 (88.5)	37 (78.7)	0.136
Thyroidectomy	85 (88.5)	42 (89.4)	1
Graves’ disease	14 (14.6)	2 (4.3)	0.09
Malignant thyroid condition	13 (13.5)	7 (14.9)	0.803
Central LND	15 (15.6)	9 (19.1)	0.637
Reoperative surgery	5 (10.6)	7 (7.3)	0.53
PTG visualised			0.066
0	5 (5.2)	2 (4.3)	
1	3 (3.1)	0	
2	11 (11.5)	11 (23.4)	
3	25 (26)	18 (38.3)	
4	52 (54.2)	16 (34)	
PTG autograft			0.223
0	77 (80.2)	33 (70.2)	
1	17 (17.7)	14 (29.8)	
2	2 (2.1)	0	
Immediate calcium substitution	13 (13.5)	2 (4.3)	0.144
Symptoms of PH	55 (57.3)	26 (55.3)	0.859
PTH POD 1, median (IQR)	11.0 (7.3–11.6)	9.1 (9–15.4)	**0.006**
PTH POD 2, median (IQR)	10.2 (7.8–13.6)	8.6 (7.8–12.2)	**0.008**
Calcium POD 1, median (IQR)	2.1 (2–1.1)	2.1 (2–2.2)	0.544
Calcium POD 2, median (IQR)	2 (1.9–2.1)	2 (1.9–2.1)	0.31

Abbreviations: LAD = lymphadenectomy, POD = postoperative day, PTG = parathyroid glands, PTH = parathyroid hormone; IQR = interquartile range.

**Table 3 jcm-11-03202-t003:** Baseline characteristics for recovery within 24 weeks compared to a later recovery.

n (%) Unless Otherwise Stated	Recovered ≤ 24 Weeks	Recovered > 24 Weeks	*p*-Value
	n = 116	n = 27	
Female gender	98 (84.5)	24 (88.9)	0.765
Thyroidectomy	104 (89.7)	23 (85.2)	0.592
Malignant thyroid condition	16 (13.8)	4 (14.8)	1
Central LAD	17 (14.7)	7 (25.9)	0.163
Graves’ disease	15 (12.9)	1 (3.7)	0.307
Reoperative surgery	8 (6.9)	4 (14.8)	0.24
PTG visualised			0.089
0	5 (4.3)	2 (7.4)	0.62
1	3 (2.6)	0	1
2	15 (12.9)	7 (25.9)	0.135
3	33 (28.4)	10 (37.0)	0.485
4	60 (51.7)	8 (29.6)	0.053
PTG autograft			1
0	90 (77.6)	20 (74.1)	
1	24 (20.7)	7 (25.9)	
2	2 (1.7)	0	
Immediate calcium substitution	14 (12.1)	1 (3.7)	0.304
Symptoms of PH	68 (58.6)	13 (48.1)	0.39
PTH POD 1, median (IQR)	10.8 (8.45–15.1)	8.8 (7.3–12.7)	**0.033**
PTH POD 2, median (IQR)	9.9 (7.78–12.53)	8.4 (7–10.45)	0.057
PTH > 15 and <65 POD 1	48 (42.1)	7 (25.9)	0.132
PTH ≥ 10 POD 1	54 (46.6)	18 (66.7)	0.086
Calcium POD 1, median (IQR)	2.1 (2–2.2)	2.0 (2–2.1)	0.096
Calcium POD 2, median (IQR)	2 (1.9–2.1)	1.95 (1.8–2.1)	0.387

Abbreviations: LAD = lymphadenectomy, POD= postoperative day, PTG = parathyroid glands, PTH = parathyroid hormone.

**Table 4 jcm-11-03202-t004:** Logistic regression analysis of potential risk factors.

	Univariable Analysis				Multivariable Analysis			
	OR	Lower CI	Upper CI	*p*-Value	OR	Lower CI	Upper CI	*p*-Value
(a) recovery within 4 weeks (within 4 weeks n = 41 vs. later than 4 weeks n = 102)								
**Graves’ disease**	3.82	1.32	11.08	**0.014**	2.40	0.72	8.07	0.156
Central LAD	0.61	0.21	1.75	0.355				
All PTG visualized	1.86	0.89	3.88	0.097	2.02	0.88	4.63	0.098
PTG autograft	1.35	0.64	2.89	0.433				
**PTH POD 1**	1.12	1.05	1.19	**<0.001**	1.13	1.06	1.20	**<0.001**
Calcium POD 1	0.40	0.03	4.70	0.464				
Symptoms	1.12	0.54	2.32	0.772				
Female gender	2.71	0.75	9.77	0.127	2.69	0.64	11.34	0.178
HL ChiSq 13.966, *p* 0.083, AUC 0.771								
(b) recovery within 24 weeks (within 12 weeks n = 116 vs. later than 24 weeks n = 27)								
Graves’ disease	3.86	0.49	30.3	0.201				
Malignity	0.92	0.28	3.01	0.890				
**All PTG visualized**	2.55	1.03	6.29	0.043	2.69	1.08	6.69	**0.033**
PTG autograft	0.92	0.38	2.25	0.856				
PTH POD 1	1.07	0.98	1.16	0.136	1.07	0.98	1.16	0.125
Ca POD 1	7.04	0.40	125.0	0.184				
Symptoms	0.65	0.28	1.52	0.325				
Female gender	0.68	0.19	2.50	0.562				
HL ChiSq 7.267, *p* 0.508, AUC 0.664								

## Data Availability

Restrictions apply to the data presented in this study. Data were obtained from the StuDoQ Thyroid Quality Assurance Registry of the German Association of Surgeons (DGAVC) and may be available to third parties only upon request to the SAVC/DGAVC and the decision at the discretion of the DGAVC.

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
