# Peer review of "How Long Does It Take to Regain Normocalcaemia in the Event of Postsurgical Hypoparathyroidism? A Detailed Time Course Analysis"

_jcm, 2022, doi:10.3390/jcm11113202_

Round 1

Reviewer 1 Report

This retrospective analysis of the time course of parathyroid gland recovery  provides a description of a large German cohort of more than 1000 thyroid resections over a one year period: 2015-1016.

The retrospective design did not permit uniform biochemical assessment of all post-surgical cases diagnosed with hypoparathyroidism.  The endocrine literature defines the  diagnosis of hypoparathyroidism as hypocalcemia with low-normal or below normal PTH levels.  The authors incorrectly state (Line 43) that no consensus exists for the definition of post-surgical hypoparathyroidism. The authors include in their PH cohort, patients who had no no biochemical profile but had evidence only of hypocalcemia symptoms or with newly prescribed calcium supplements.  Symptoms of hypocalcemia may occur with a drop in PTH level but with no evidence of hypocalcemia or below normal serum PTH levels. The authors state that calcium supplements are routinely provided to post-op patients (line 245). The authors’ rather loose definition led to an over-estimation of the numbers of post-surgical hypopara cases and thus also the rate of recovery.  Their inclusion/exclusion criteria do not capture the true numbers who actually had the disease, however transient it might have been.   

Section 2.2 line 90:  It is unclear how long “the detailed follow-up program” continued. At what point did the investigators cease monthly follow-up calls—at 12 months? The article currently says that “follow-up was only terminated if a full recovery had been validated...”

Conventional therapy for hypoparathyroidism is active vitamin D and calcium supplements.  Did you track active vitamin D therapy (alphacalcidol or calcitriol) or just calcium supplements?

Line 48 I recommend referring to PH as a hormonal insufficiency state.

These cases  were all post thyroidectomies? No other neck surgeries?

The Discussion is difficult to follow  and contains redundant statements. I recommend editing this section and refer only to studies that are relevant.  For example, Cusano (first author) et al (ref#6) studied surgery of patients with hyperparathyroidism. This is a different disease, and such patients were excluded from this cohort.

Reviewer 2 Report

The authors aimed to perform a detailed analysis of the biochemical recovery of patients with PH and to detect the predictors for early recovery. 

The topic is clinically meaningful, postsurgical hypoparathyroidism is the most common complication of thyroidectomy, even in high volume centers. The article is well written, the design of the study is appropriate, the number of patients included in the study is relevant.  

Minor spelling check is required. (eg. R122: 'reimplantation' instead of 'replantation')

Minor comments/suggestions:

Please define low PTH and calcium values, as well as recovery - in the material and methods section (the variables are categorized in the results section but it is not explained in methods), as it was observed that the prevalence of PH varies with its definition. 

If data is available, is there a correlation/concomitance of HP with other complications of thyroid surgery?

Conclusions should be very clearly stated. The last phrase is not clear, please rephrase: R398 "While barely reduced postsurgical levels of PTH evolved as a predictor of early recovery a later recovery, even after a long observation period, became more likely in particular with visualization of all PTGs during surgery"
